A retrospective study on the outcomes of cataract surgery in an Eastern Regional Health Authority hospital of Trinidad and Tobago

Sonron Ebiakpo-aboere 1
Tripathi Vrijesh 1 vrijesh.tripathi@sta.uwi.edu
Bridgemohan Petra 2
Sharma Subash 3
1 Department of Mathematics & Statistics, Faculty of Science and Technology, The University of the West Indies , St. Augustine , Trinidad and Tobago
2 Ophthalmology Department, Sangre Grande Hospital, Eastern Regional Health Authority , Trinidad and Tobago
3 School of Optometry and Visual Sciences, The University of the West Indies , St. Augustine , Trinidad and Tobago
Lo Presti Alessandra
Electronic publication date: 2015 Sep 3
Publication date: 2015
Volume: 3
Electronic Location ID: e1222
Received 2015 Jun 10; Accepted 2015 Aug 9
Copyright: © 2015 Sonron et al.
Copyright year: 2015
Copyright holder: Sonron et al.
License: This is an open access article distributed under the terms of the Creative Commons Attribution License, which permits unrestricted use, distribution, reproduction and adaptation in any medium and for any purpose provided that it is properly attributed. For attribution, the original author(s), title, publication source (PeerJ) and either DOI or URL of the article must be cited.
License URL: https://creativecommons.org/licenses/by/4.0/

Keywords: Visual outcomes, Logistic regression, Cataract surgery, Ocular co-morbidity, Cataract surgery

Funding: The authors received no funding for this work.

==============================
Background. Worldwide, cataract is a major cause of blindness. The paper aims to evaluate factors associated with borderline and poor outcomes of cataract surgery at an Eastern Regional Health Authority (ERHA) hospital in Trinidad and Tobago.

Materials and Methods. A hospital-based, retrospective study was done on 401 patients who had undergone cataract surgery (unilateral and bilateral) at an ERHA Hospital between March 2009 and September 2014. Data was collected on variables concerning demographic, medical history, surgical history, ocular findings and visual acuity (VA). The outcome variable of interest was Snellen’s post-operative (presenting) VA which was transformed into a dichotomous variable with borderline and poor outcomes as one and good outcomes as the other. Data were analysed using univariate and multivariate logistic regression analyses.

Results. Good outcome (presenting VA 6/18 or better) was seen in 350 (67%) eyes. The fitted model consisted of ocular co-morbidity (OR =2.133; 95% CI [1.346–3.380]), hypertension (OR = 0.520; 95% CI [0.381–0.928]), surgical procedure (OR = 1.56; 95% CI [1.004–2.425]), good preoperative VA (OR = 0.388, 95% CI [0.211–0.714]), borderline preoperative VA (OR = 0.485; 95% CI = [0.278–0.843]) and year of first visit to clinic (OR = 2.243; 95% CI [1.215–4.141]).

Conclusion. There is a need for community-based outreach to increase awareness of eye health and diseases. It is recommended that the general population is encouraged to take responsibility for personal management. The facilities at the Hospital should also be enhanced.

Introduction

Cataract is defined as the opacification of the eye’s natural lens and can develop in a few months or take several years (Baltussen, Sylla & Mariotti, 2004; Fattore & Torbica, 2008). Global data on visual impairment attributes 51% of blindness in the world to cataract affecting approximately 18 million people worldwide (International Agency for the Prevention of Blindness, 2015; Pascolini & Mariotti, 2012; Lindfield et al., 2012). In the Caribbean, non-operated cataract is the most prevalent cause of blindness (PAHO, 2010). According to the Barbados Eye Studies, visual impairment occurs in 12% of the people 40–84 years old and 3% suffer from severe visual impairment. Furthermore, age-related cataract alongside open-angle glaucoma (OAG) accounts for 73.2% of blindness, with two-thirds of low vision (visual acuity from <6/18 to <6/120) being attributed to cataract (Leske et al., 2010). Approximately 70% of eye surgeries in Trinidad and Tobago are for the removal of cataracts with 2,500 cataract extractions and presentation of 3,000 new cases annually at the public hospitals (Choy, 2012). The risk factors for cataract development include gender, diabetes mellitus, exposure to excessive sunlight, life style, eye injury and use of steroids (The Royal College of Ophthalmologists, 2010). Data from twin genetic studies estimate a heritability between 48% and 59% of age-related cataract (Hammond et al., 2000; Hammond et al., 2001).

The Ophthalmology Clinic at the Sangre Grande (SG) Hospital is administrated and managed by the Eastern Regional Health Authority (ERHA) of Trinidad and Tobago. The ERHA provides healthcare for the catchment population of approximately 120,000 from Matelot in the north to Guayaguayere, Rio Claro and Brothers Road in the South to Valencia in the east. Cataract surgical services offered by the Ophthalmology Clinic began in March 2009 and are done as outpatient surgeries. The clinic is a referral centre and cataract patients are routinely given an appointment date that is 6 months from the time they come in with their referral. Those who suffer from complications such as glaucoma and diabetic retinopathy (DR) are given priority. This results in some patients being on the waiting list for at least 2 years.

The outcomes of cataract surgery can be measured using an objective clinical indicator such as visual acuity (VA) and/or subjective indicators such as quality of life (QOL) and visual functioning (VF) instruments. This paper uses VA to measure the outcomes of cataract surgery at SG Hospital. In its guidelines, the World Health Organisation (WHO) recommends that all patients undergoing cataract surgery should have VA measured in each eye preoperatively, and anytime between discharge and 12 weeks (World Health Organization, 1998). The objective of this paper is to evaluate the visual outcomes at the SG Hospital and to investigate factors associated with borderline and poor outcomes. These indentified factors can possibly lead to the establishment of a framework for improving the outcomes of cataract surgery.

Materials and Methods

This was a hospital-based, retrospective study on patients who had undergone cataract surgery, unilateral or bilateral, at the Sangre Grande Hospital of the ERHA from March 2009 to September 2014. The main types of cataract surgery done at SG Hospital are phacoemulsification (Phaco) and extracapsular cataract extraction (ECCE) which are both conducted with intraocular lens (IOL) insertion. There were a total of 1,100 cataract surgeries that had taken place at SG Hospital during this time period. Patients were selected randomly from the database maintained by the Medical Records Unit and their medical files were reviewed if physically available at the time of the study. This led to a selection of 277 patients with unilateral and 124 patients with bilateral cataract surgery. Around 1% of the surgeries performed were Small Incision Cataract surgery (SICS) and these cases were excluded from univariate and multivariate analyses because of their low number. To avoid duplication of patient’s information in cases of bilateral cataract surgery, a random number table was used to select which eye would be included in the regression analysis. This led to the selection of 401 patients aged 18–100 as the sample for the study.

Explanatory variables

The following variables were recorded from their files: demographic variables such as sex, date of birth, age, marital status, occupation, residence; other variables such as year of first visit to the clinic, referral source, and time between visit to clinic and cataract surgery; ocular findings such as first measurement of VA, post-operative (presenting) VA and intraocular pressure (IOP); ocular co-morbidities including glaucoma, DR and age-related macular degeneration (ARMD); presence of co-incident diseases such as diabetes mellitus, hypertension, cardiovascular disease and high cholesterol; and, variables on surgical history such as surgical procedure used, and previous surgeries, if any.

Statistical technique

The outcome variable of interest was Snellen’s post-operative VA measured at the follow up starting from at least 3 months after surgery. This variable was transformed into a dichotomous variable (Good and Poor) based on the measurements of post-operative VA with borderline and poor outcomes grouped as poor. Univariate logistic regression analysis was conducted to explore the association of the co-variates and the dependent variable. The variables that were moderately significant (<0.10) were then included in forward step-wise multivariate logistic regression. MS Excel, R and IBM SPSS 20.0 statistical packages were used for data management and analyses.

Ethical approval

The University of the West Indies (UWI) St. Augustine Campus Ethics Committee provided the written approval by letter dated 22 January, 2015 and the Research Ethics Committee of the ERHA provided written ethical approval through letter reference PHO:04/15 dated 25 February 2015.

Results

Comparison of visual acuity pre and post cataract surgery for data on 525 eyes of 401 patients is graphically presented in Fig. 1. Upon first visit to the clinic, preoperative VA showed that 114 eyes (22%) had good VA, 140 eyes (27%) had borderline VA and 271 eyes (51%) had poor VA. Post-operative cataract VA resulted in 350 eyes (67%) having good VA, 110 eyes (21%) borderline VA and 65 eyes (12%) poor VA. This shows that visual acuity improved in 326 eyes (62%), but worsened in 33 eyes (6%) with no change in 166 eyes (32%).

Figure 1 Visual acuity categories before and after cataract surgery.

Socio-demographic characteristics

The number, frequency and univariate analyses of socio-demographic characteristics of the sample population are given in Table 1. The sample of 401 adults consisted of 200 (49.9%) men and 201 (50.1%) women, who had undergone cataract surgery in one (277 subjects) or both eyes (124 subjects). The age at surgery ranged from 23 to 98 years (mean 68.1 ± 12.1 years). The age at surgery was calculated from having recorded the date of birth and date of surgery for all subjects. Most patients were within the 51–80 age group (76.8%). One hundred and sixty-four (40.9%) patients were retired. The patients mainly resided in areas that are administered by ERHA with 246 (61.3%) patients living in areas including Sangre Grande, Manzanilla and Toco. Referral sources came largely from health centres within the ERHA (34.4%) and optical centres such as Ferreira Ltd and Dalton Brown & Long (31.1%). The odds of having a poor outcome were increased by being male (OR = 1.37, 95% CI [0.91–2.08]); retired (OR = 1.41, 95% CI [0.85–2.34]), unemployed or on disability allowance (OR = 2.58, 95% CI [0.87–7.67]). However, only the variable year of first visit to clinic (OR = 1.85, 95% CI [1.07–3.20]) was statistically significant at p < 0.05 level.

Table 1 Univariate analyses of socio-demographic factors of patients based on visual outcomes after cataract surgery.

	Visual outcome				
Variables	Good n = 263 (66%)	Poor n = 138 (34%)	Total	OR & 95% CI for OR	P-value	
Age, years	67.70 ± 11.43	68.82 ± 13.28	68.01 ± 12.09	1.008 (0.99–1.03)	0.378	
Age group, years						
≤50	20 (7.6)	13 (9.4)	33	1		
51–80	208 (79.1)	100 (72.5)	308	0.74 (0.35–1.55)	0.423	
>80	35 (13.3)	25 (18.1)	60	1.10 (0.46–2.61)	0.831	
Sex						
Female	139 (52.9)	62 (44.9)	201	1		
Male	124 (47.1)	76 (55.1)	200	1.37 (0.91–2.08)	0.132	
Marital status						
Married	125 (47.5)	54 (39.1)	179	1	0.175	
Unmarried	127 (48.3)	80 (38.6)	207	1.46 (0.95–2.23)	0.082	
Not stated	11 (4.2)	4 (2.9)	15	0.84 (0.26–2.76)	0.776	
Year of first visit to clinic						
2009–2013	229 (87.4)	109 (79.0)	338	1		
2005–2008	33 (12.6)	29 (21.0)	62	1.85 (1.07–3.20)	0.028*	
Time between first visit and operation, years	1.55 ± 1.48	1.71 ± 1.49	1.60 ± 1.47	1.08 (0.79–1.48)	0.638	
Time between first visit and operation, years						
<= 2	188 (71.5)	90 (71.5)	278	1		
>2	75 (28.5)	48 (34.8)	123	1.34 (0.86–2.08)	0.197	
Residence						
ERHA areas	165 (62.7)	81 (58.7)	246	1		
Other	98 (37.3)	57 (41.3)	155	1.19 (0.78–1.81)	0.430	
Occupation						
Employed	79 (30.0)	35 (25.4)	114	1		
Retired	101 (38.4)	63 (45.7)	164	1.41 (0.85–2.34)	0.186	
Unemployed/disability allowance	7 (2.7)	8 (5.8)	15	2.58 (0.87–7.67)	0.088	
Not stated	76 (28.9)	32 (23.2)	108	0.95 (0.54–1.69)	0.862	
Referral sources						
ERHA	90 (34.2)	48 (34.8)	138	1		
Optical	84 (31.9)	41 (29.7)	125	0.92 (0.55–1.53)	0.734	
Other	69 (26.2)	38 (27.5)	107	1.03 (0.61–1.75)	0.905	
Not stated	20 (7.6)	11 (8.0)	31	1.03 (0.46–2.33)	0.941	
Notes.

* p ≤ 0.05.

Medical and surgical history

An examination of medical history shows that diabetes mellitus was present in 41% (n = 163), while hypertension was present in 45% (n = 182) of the patients. There were more women than men who had diabetes mellitus and hypertension. Patients were also suffering from cardiovascular disease (9%) and high cholesterol (11%). Ocular co-morbidity was present in 32% (n = 128) of the patients. Distribution and the results of univariate analyses of each variable are presented in Table 2.

Table 2 Univariate analyses of clinical and surgical factors of patients based on visual outcome after cataract surgery.

	Visual outcome				
Variables	Good n = 263 (66%)	Poor n = 138 (34%)	Total	OR & 95% CI for OR	p-value	
Preoperative VA						
Poor	122 (46.4)	88 (63.8)	210	1		
Good	66 (25.1)	20 (14.5)	86	0.42 (0.24–0.74)	0.003**	
Borderline	75 (28.5)	30 (21.7)	105	0.56 (0.34–0.92)	0.022*	
Ocular Co-Morbiditya						
Absent	195 (74.1)	79 (57.2)	274	1	–	
Present	68 (25.9)	59 (42.8)	127	2.14 (1.39–3.31)	0.001**	
Diabetes						
Absent	164 (62.4)	74 (53.6)	238	1	–	
Present	99 (37.6)	64 (46.4)	163	1.43 (0.94–2.18)	0.091	
Hypertension						
Absent	134 (51.0)	85 (61.6)	219	1	–	
Present	129 (49.0)	53 (38.4)	182	0.65 (0.43–0.99)	0.043*	
Cardiovascular disease						
Absent	236 (89.7)	127 (92.0)	363	1	–	
Present	27 (10.3)	11 (8.0)	38	0.76 (0.36–1.58)	0.457	
High cholesterol						
Absent	236 (89.7)	119 (86.2)	355	1	–	
Present	27 (10.3)	19 (13.8)	46	1.40 (0.75–2.61)	0.297	
Intraocular pressure						
Normal	238 (90.8)	112 (81.2)	350	1	–	
High	24 (9.2)	26 (18.8)	50	2.30 (1.27–4.19)	0.006**	
Eye surgery done on						
Right	146 (55.5)	73 (52.9)	219	1	–	
Left	117 (44.5)	65 (47.1)	182	1.11 (0.74–1.68)	0.617	
Surgical procedure						
PHACO	158 (60.3)	64 (47.1)	222	1	–	
ECCE	104 (39.7)	72 (52.9)	176	1.71 (1.13–2.60)	0.012*	
Other eye surgeries						
No	249 (94.7)	120 (87.0)	369	1	–	
Yesb	14 (5.3)	18 (13.0)	32	2.67 (1.28–5.55)	0.009**	
Notes.

* p ≤ 0.05.

** p < 0.01.

a includes mainly glaucoma, diabetic retinopathy, age related macular degeneration and uveitis.

b includes anterior vitrectomy, trabeculectomy and pterygium.

Univariate logistic regression found the following six clinical variables statistically significant (p < 0.05): good pre-operative VA (OR = 0.42, 95% CI [0.24–0.74]), borderline pre-operative VA (OR = 0.56, 95% CI [0.34–0.92]), ocular co-morbidity (OR = 2.14, 95% CI [1.39–3.31]), hypertension (OR = 0.65, 95% CI [0.43–0.99]), intraocular pressure (OR = 2.30, 95% CI [1.27–4.19]), surgical procedure (OR = 1.71, 95% CI [1.13–2.60]) and other eye surgeries excluding cataract (OR = 2.67, 95% CI [1.28–5.55]). The odds of having poor outcome increased with diabetes mellitus (OR = 1.43, 95% CI [0.94–2.18]); high cholesterol (OR = 1.40, 95% CI [0.75–2.61]); and having cataract surgery on the left eye (OR = 1.11, 95% CI [0.74–1.68]). However, these were statistically non-significant in our analyses. Surgical technique used was statistcally significant with Phacoemulsification being better than ECCE (OR = 1.71, 95% CI [1.13–2.60]).

Multivariate analyses

Multivariate logistic regression showed that other eye surgeries and intraocular pressure did not add significantly to the model. Table 3 shows the variables of the fitted model with their p-values, odds ratios and 95% confidence intervals. The fitted model shows that there is an inverse relationship of poor outcomes with hypertension, good and borderline preoperative VA. The odds of having a poor outcome for those with hypertension was 40% less (OR = 0.60, 95% CI [0.38–0.94]) than those with no hypertension. The odds of having a poor outcome were reduced by 61% in those with good preoperative VA (OR = 0.39, 95% CI [0.21–0.72]) and by 51% in those with borderline preoperative VA (OR = 0.49, 95% CI [0.28– 0.86]) when compared to those measured with poor preoperative VA. On the other hand, there is a positive relationship of poor outcomes with ocular co-morbidity, surgical procedure and the year the patient first visited the clinic. Those with an ocular co-morbidity have two-fold increased odds of a poor outcome compared with those with no ocular co-morbidity present (OR = 2.13, 95% CI [1.36–3.42]); year of first visit to the clinic prior to 2009 resulted in twice the odds of a poor outcome compared with those who first visited from 2009 till 2014 (OR = 2.24, 95% CI [1.20–4.10]); and, the odds of a poor outcome were increased by 56% for those patients whose surgical procedure was ECCE (OR = 1.56, 95% CI [1.01–2.45]).

Table 3 Multivariate analyses of factors affecting visual outcome after cataract surgery.

Variables	Coff (β)	OR & 95% CI for OR	p-value	
Hypertension	−0.52	0.60 (0.38–0.94)	0.022*	
Ocular co-morbidity	0.76	2.133 (1.36–3.42)	0.001**	
Good preoperative VA	−0.95	0.388 (0.21–0.72)	0.002**	
Borderline preoperative VA	−0.72	0.485 (0.28–0.86)	0.010*	
Year of first visit	0.81	2.243 (1.20–4.10)	0.010*	
Surgical procedure	0.45	1.560 (1.013–2.45)	0.048*	
Constant	−0.66	0.52	0.002**	
Notes.

* p ≤ 0.05.

** p < 0.01.

Discussion

The WHO recommends that a clinical audit record be maintained for all cases of cataract surgery. It mandates that with available correction, acceptable visual outcomes post cataract surgery are good outcome (6/18 or better) in >80% of cases, borderline outcome (<6/18 to 6/60) in <15%, and poor outcome (<6/60) in <5% of cases (Pararajasegaram, 2002).

Our data set did not meet these criterias with the figures falling short in each of the categories. The results from population-based surveys elsewhere have shown that 40% or greater of postoperative eyes have a presenting VA of worse than 6/18 (Dandona et al., 1999; He et al., 1999; Murthy et al., 2001). In this study, the corresponding percentage was just under 34% which is still significantly high when compared to the benchmark of less than 20% recommended by World Health Organization (1998). It is important that both awareness and facilities are upgraded for the required population.

This study identifies factors that are associated with borderline and poor outcomes of cataract surgery performed at SG Hospital in Trinidad and Tobago. While there is little availability of literature concerning studies done locally or in the Caribbean region, the results of this study are comparable with that of previous international studies (Desai, Minassian & Reidy, 1999; Norregaard et al., 1998). Some studies (Gupta et al., 2013; Norregaard et al., 1998) use the best corrected visual outcome (BCVA) when assessing visual outcomes but this study used post-operative (presenting) VA since BCVA was not available for all patients. This does not nullify the results of the study as “it is the presenting vision that represents the actual circumstances under which people function in day-to-day activities” (Leon, 2000).

Demographic factors such as age, sex and place of residence are not significant factors in this study. Age has been identified as significantly associated with poorer visual outcomes (Norregaard et al., 1998), with patients aged 90 years and over having four times the risk of poor visual outcomes when compared to those aged 50 to 59 years (Desai, Minassian & Reidy, 1999). There are equal number of males and females in our sample for this study and males have higher odds of poor outcomes when compared to females. However, results from a Pakistan survey and a study from Rajasthan in India show that females are associated with poor outcomes (Bourne et al., 2007; Murthy et al., 2001). Place of residence has limited effect on visual outcomes. Some studies report poor outcomes among rural patients (Dandona et al., 1999), while others ascribe no significant association (Nirmalan et al., 2002). Comparison of places of residences (rural to urban) was beyond the scope of this study as it was hospital-based. For this study, those who did not reside in ERHA regions had increased odds of poor outcomes. It is notable that most patients appeared to wait until vision was very poor before visiting the clinic. One plausible reason could be that due to the slow, progressive decline in vision that characterizes the development of cataracts, patients are not aware earlier of the decrease in visual function (Bellan et al., 2008). There is no supporting literature to explain why the odds of a poor outcome increased if the patient first visited the clinic prior to 2009. The only plausible explanation is that those who first visited the clinic prior to the opening of the Ophthalmology Clinic in 2009 are more likely to have had a longer waiting period for cataract surgery. This longer waiting period could possibly have led to higher risks for development of further complications, worsening of symptoms or poor prognosis and outcome following surgery (Hadjistavropoulos, Snider & Bartlett, 1998; Mojon-Azzi & Mojon, 2007).

The presence of an ocular co-morbidity increases the likelihood of a poor outcome by at least two-fold. This is consistent with the results of previous studies (Desai, Minassian & Reidy, 1999; Kshitiz, Gupta & Dhaliwal, 2012; Norregaard et al., 1998) and is seen as a frequent reason for poor self-assessed outcomes of cataract extraction (Lundström, Stenevi & Thorburn, 1999; Rönbeck, Lundström & Kugelberg, 2011). However, there is no consensus on a proven association between cataract surgery and ocular diseases such as glaucoma (Bernth-Petersen & Bach, 1983; Harding, Harding & Egerton, 1989; Heltzer, 2004; Kung et al., 2015), DR (World Health Organization, 2006; Hooper et al., 2012; Hong et al., 2009) and ARMD (Bellan et al., 2008; Chang et al., 2011; Velez & Weiter, 2002).

Despite the association between ocular co-morbidity and cataract surgery, Lundström et al. (2012) state that it is not a contraindication for cataract surgery. Analyses of systemic co-morbidities such as diabetes mellitus, hypertension, cardiovascular disease and high cholesterol revealed that only hypertension is significantly associated with poor visual outcomes. Several studies identify diabetes mellitus as a risk factor for cataract (Harding et al., 1993; Leske et al., 2010). Although diabetes mellitus is not significantly associated with poor outcomes in this study, univariate analyses showed evidence that the odds of poor outcome are increased on having diabetes mellitus. Cardiovascular disease is shown not to be significantly associated with poor outcomes in this study. This is supported by Tan, Wang & Mitchell (2008) though other studies support the association of the cardiovascular disease and cataract (Goodrich et al., 1999). Hypertension was inversely related to poor visual outcomes as higher levels reduced the risk of poor visual outcomes when compared to those with no hypertension. There is little literature to explain what leads hypertension to be associated with cataract but Lee et al. (1997) in their study imply that hypertension gives rise to conformational changes in the lens capsule.

Phacoemulsification, a procedure developed by Charles D. Kelman (Goldstein, 2004), is the preferred surgical procedure while ECCE is performed depending on the condition of the anterior chamber, iris, and lens (Lundström et al., 2012). Phacoemulsification requires a smaller incision and is suture-less which can lead to significant reduction in surgically-induced astigmatism (Kshitiz, Gupta & Dhaliwal, 2012). In this study, the odds of a poor outcome are increased when the surgical procedure was ECCE which is consistent with previous studies (Desai, Minassian & Reidy, 1999; Minassian et al., 2001).

The study shows that the odds of a poor outcome increase from borderline to poor preoperative VA. This is consistent with the International Cataract Surgery Outcomes Study, where poor preoperative VA is highlighted as one of the predictors of a poor visual outcome (Norregaard, 2007).

There are several limitations in this study. Firstly, this is a retrospective hospital-based study. Secondly, BCVA measurements were not recorded in the data readily available in the medical records. Despite these limitations, it is believed that this study adequately identifies factors that are associated with borderline and poor visual outcomes of cataract surgery.

Conclusion

This study identifies the factors that significantly affect visual outcomes such as surgical procedure, preoperative VA, hypertension and ocular co-morbidities. It is recommended that the phacoemulsification technique be increasingly utilized due to evidence for better cataract surgery outcomes when compared to ECCE. There is also a need for community-based outreach to increase awareness of the importance of eye health. Other recommendations include implementing the use of a quality of life (QOL) instrument to assess outcomes of cataract surgery alongside the clinical indicator of visual acuity. It is also recommended that all attempts are made to ensure that the medical records of patients are complete and thorough as much as possible.

The authors would like to thank the staff of the Medical Records Unit of the Sangre Grande Hospital, for their help in providing information from the database and locating medical records used for this study.

Additional Information and Declarations

Competing Interests

Author Contributions

Human Ethics

Data Availability

The authors declare there are no competing interests.

Ebiakpo-aboere Sonron conceived and designed the experiments, analyzed the data, wrote the paper, prepared figures and/or tables, reviewed drafts of the paper.

Vrijesh Tripathi conceived and designed the experiments, analyzed the data, wrote the paper, reviewed drafts of the paper.

Petra Bridgemohan and Subash Sharma conceived and designed the experiments, reviewed drafts of the paper.

The following information was supplied relating to ethical approvals (i.e., approving body and any reference numbers):

The University of the West Indies (UWI) St. Augustine Campus Ethics Committee provided the written approval by letter dated 22 January, 2015 and the Research Ethics Committee of the ERHA provided written ethical approval through letter reference PHO:04/15 dated 25 February 2015.

The following information was supplied regarding the availability of data:

Raw data was accessed from medical records of patients of cataract surgery maintained by the Eastern Regional Health Authority. To access this data, future researchers can contact the Eastern Regional Health Authority Head Office, Sangre Grande, Trinidad: http://www.health.gov.tt/sitepages/default.aspx?id=90.

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
