# Peer review of "A retrospective study on the outcomes of cataract surgery in an Eastern Regional Health Authority hospital of Trinidad and Tobago"

_PeerJ, doi:10.7717/peerj.1222_

## Round 0.1 · original submission · Minor Revisions

In the manuscript entitled "A retrospective study on the outcomes of cataract surgery in an Eastern Regional Health Authority Hospital of Trinidad and Tobago" the authors evaluated the visual outcomes at the SG Hospital and investigated factors associated with borderline and poor outcomes. The study is well designed but the authors have to perform some minor revisions following the indication of the Reviewers to have a better precision in both method and result section.

Reviewer 1 ·

Basic reporting

No Comments

Experimental design

No Comments

Validity of the findings

No Comments

Additional comments

Sonron et al., performed a retrospective study to evaluate factors associated with borderline and poor outcomes of cataract surgery at an Eastern Regional Health Authority (ERHA) hospital in Trinidad and Tobago from March 2009 to September 2014.
The authors performed a good analysis analysing mostly of the possible variable to indetify the possible association. Good statistical analysis.

Comments:
- Results section line 135. There is the description of the 525 eyes operated but the sample included 401 adults. It is not clear the real sample analysed. Please write better.
- The resolution of the figure 1 is low. Please increase it.

Reviewer 2 ·

Basic reporting

"No Comments".

Experimental design

"No Comments".

Validity of the findings

"No Comments".

Additional comments

Sonron et al., in this paper try to evaluate the visual outcomes at the SG Hospital in Trinidad and Tobago and to investigate factors associated with borderline and poor outcomes. 


Comments:
- I suggest the Authors to use an English more appropriate.
- the introduction section is in my opinion too long, the authors start in this section a very long discussion about the general System and the clinic curing the Cataract disturbs in Trinidad and Tobago. I suggest them to reduce this part and to focus their attention only on the aim of their study.
- In the Statistical Technique section there isn’t in my opinion a detailed description about the method and the program used to conduct this kind of analysis. I suggest them to write more in detail this part.
-In the result section they talk about 525 eyes operated , but in the method section they said to have enrolled 401 patients. Why they decide to speak about the eyes and not the patients? In my opinion this part is just a little bit confused and in this way it is not easy understand the real denominator analyzed. I suggest to them to rewrite better this part.

-The discussion section is in my opinion long and ridondant in some part, I suggest to them just to reduce a little bit this section.


Despite this considerations the paper is in my opinion readable, I think that try to evaluate the visual outcomes at the SG Hospital in Trinidad and Tobago and try to investigate factors associated with borderline and poor outcomes could be important because these indentified factors could possibly lead to the establishment of 
a framework for improving the outcomes of cataract surgery.

Reviewer 3 ·

Basic reporting

Manuscript is generally written in good english, some sentences are too long and they reduce the overall readability of the manuscript.
Page 11 line 131 the word "characterises" should be "characterizes".

Experimental design

No Comments

Validity of the findings

Methods: Authors should clearly described in the methods the number of eyes analyzed explaining the inclusion and exclusion criteria respect to the number of patients.

Additional comments

The manuscript here presented describes the outcomes of cataract surgery in an Eastern Regional Health Authority Hospital of Trinidad and Tobago. The retrospective study is well designed and results generally well described. Only one critical point is not clear. In the methods the number of the eyes considered and whether in case of bilateral cataract surgery both eyes are included or not is not well explained. Material and Methods state "For those patients who had undergone bilateral cataract surgery, a random number table was used to select which eye would be included in the regression analysis" (page 6 lines 117-118). This sentence gives the impression that one eye (randomly chosen) has been analyzed for each patient, therefore 401 as the number of the patients (page 5 line 107). However, in the results the total number of operated eyes described is 525 (page 7 line 135). Even if both eyes are considered in case of bilateral cataract surgery (124 patient; page 8 line 146) the total number should be 463. Why there is this discrepancy?

---

## Round 0.2 · accepted · Accept

The authors have improved the manuscript, performing the revision required according to the suggestion of the Reviewers. There are not additional comments or revision that need to be done.

Reviewer 3 ·

Basic reporting

No Comments

Experimental design

No Comments

Validity of the findings

No Comments

Additional comments

Authors have fully provided explanation for uncertain issues and the overall manuscript has improved its readability.